# Study of low-dose PET image recovery using supervised learning with CycleGAN

Kui Zhao[1]☯, Long Zhou[2,3]☯, Size Gao[3], Xiaozhuang Wang[2,3], Yaofa Wang[2,3], Xin Zhao[1], Huatao Wang[1], Kanfeng Liu[1], Yunqi Zhu[1], Hongwei Ye[2,3]*

1 Department of PET Center, The First Affiliated Hospital, College of Medicine, Zhejiang University, Hangzhou, China, 2 Zhejiang Minfound Intelligent Healthcare Technology Co., Ltd., Hangzhou, China, 3 MinFound Medical Systems Co., Ltd., China

☯ These authors contributed equally to this work.
* Hongwei.Ye@minfound.com

**Data Availability Statement:** The data are available within the paper and on Figshare: https://figshare.com/articles/dataset/LDPET/12686099.

**Funding:** MinFound Medical Systems Co., Ltd provided support for this study in the form of salaries for authors HY, LZ, SG, XW, and YW. The

## Abstract

PET is a popular medical imaging modality for various clinical applications, including diagnosis and image-guided radiation therapy. The low-dose PET (LDPET) at a minimized radiation dosage is highly desirable in clinic since PET imaging involves ionizing radiation, and raises concerns about the risk of radiation exposure. However, the reduced dose of radioactive tracers could impact the image quality and clinical diagnosis. In this paper, a supervised deep learning approach with a generative adversarial network (GAN) and the cycle-consistency loss, Wasserstein distance loss, and an additional supervised learning loss, named as S-CycleGAN, is proposed to establish a non-linear end-to-end mapping model, and used to recover LDPET brain images. The proposed model, and two recently-published deep learning methods (RED-CNN and 3D-cGAN) were applied to 10% and 30% dose of 10 testing datasets, and a series of simulation datasets embedded lesions with different activities, sizes, and shapes. Besides vision comparisons, six measures including the NRMSE, SSIM, PSNR, LPIPS, $SUV_{max}$ and $SUV_{mean}$ were evaluated for 10 testing datasets and 45 simulated datasets. Our S-CycleGAN approach had comparable SSIM and PSNR, slightly higher noise but a better perception score and preserving image details, much better $SUV_{mean}$ and $SUV_{max}$, as compared to RED-CNN and 3D-cGAN. Quantitative and qualitative evaluations indicate the proposed approach is accurate, efficient and robust as compared to other state-of-the-art deep learning methods.

## Introduction

Positron Emission Tomography (PET) is a widely used imaging modality for various clinical applications, such as lesion malignancy, disease stage, and treatment monitoring [1–3]. Compared with computed tomography (CT) and magnetic resonance imaging (MRI), PET is a functional imaging technique that detects the metabolism processes of human body [4]. To reach a certain PET image quality for diagnostic purposes, a typical dose of injected radioactive tracers usually ranges from 185~555 MBq, depending on PET scanners, protocols,

specific roles of these authors are articulated in the 'author contributions' section. The funders played a role in study design, data collection and analysis, decision to publish, or preparation of the manuscript. No additional external funding was received for this study.

**Competing interests:** The authors have read the journal's policy and have the following competing interests: HY, LZ, SG, XW, and YW are paid employees of MinFound Medical Systems Co., Ltd. There are no patents, products in development or marketed products associated with this research to declare. This does not alter our adherence to PLOS ONE policies on sharing data and materials.

reconstruction methods, patients and so on. Since high gamma radiation dosage in a patient may induce genetic damages and cancerous diseases [5–7], it inevitably raises concerns about the potential higher risk of radiation exposure damage. Thus, it is desirable to reduce the dose of radioactive tracers in PET imaging. However, the major drawback of dose reduction is that higher noise, worse contrast and information loss may be involved in the reconstructed images, resulting in an inferior image quality and unreliable diagnosis.

A series of methods has been proposed to improve the image quality for the low-dose PET (LDPET) imaging, while preserving crucial diagnosis information. Those algorithms can be roughly categorized into traditional methods such as iterative reconstruction algorithms [8, 9], post-processing methods [10–13] and deep learning based methods [14–22]. In general, those strategies for improving PET image quality are either hardware-oriented or computationally intensive. Besides, the LDPET image contains more complex spatial variations, correlations and statistical noise than the full-dose PET (FDPET) image, which limits the performance of the traditional methods.

Recently, deep learning has drawn a mount of attention in computer vision applications and medical image analysis areas [4–7, 23]. For instance, the image classification [24] and face verification [25] can achieve human-level performance. Algorithms based on deep learning have made some success in low-dose CT (LDCT) reconstruction and denoising [14–18]. These methods learn a non-linear mapping from a LDCT image to high-quality CT image to recover missing high-frequency details. While in recovering or denoising LDPET, there are much fewer works with deep learning methods reported. Xiang et al. [19] proposed a deep auto-context CNN model that synthesized a high quality image from 1/4 of FDPET image and corresponding MR T1-image. Xu et al. [20] used a U-Net [26] like network to recover a full-dose quality PET image from 1/200 of FDPET image, and applied a multi-slice input strategy to make the network more robust to noise. Wang et al. [21] designed an end-to-end framework based on 3D conditional GANs (3D-cGANs) to estimate the high-quality PET image from the corresponding LDPET image. The 3D convolution operation makes the model avoid the discontinuous cross-like artifacts that usually occurs in 2D convolution based models. Kaplan et al. [22] proposed a deep learning model that takes specific image features into account in the loss function to denoise 1/10 of FDPET image. Chen et al. [27] proposed to combine both PET and MR information to synthesize high quality and accurate PET images. More recent work from Ouyang et al. [28] suggests that combining a generative adversarial network (GAN) with feature matching into the discriminator can lead to similar performance even without the MR information.

Rather than using deep learning method as a post-processing tool, Gong et al. [29] proposed a residual convolutional auto-encoder within a Machine Learning framework to denoise PET images. More recently, Haggstrom et al. [30] took the PET sinogram data as the input and directly generate PET reconstructed images, highlighting a 100-fold speedup for reconstruction compared to standard iterative techniques such as ordered subset expectation maximization (OSEM).

In general, physicians use both the maximum SUV ($SUV_{max}$) and the mean SUV ($SUV_{mean}$) to characterize the high uptakes [31], but $SUV_{max}$ is more often used in practice since $SUV_{mean}$ heavily depends on volume of interest (VOI) selected, while $SUV_{max}$ value is unique and reproducible in VOI [32, 33]. Inspired by the most recent advanced neural networks, such as Dense-Net [34], Residual CNN [35], and CycleGAN [36], a cycle Wasserstein regression adversarial training framework, named S-CycleGAN, is proposed and studied for the PET brain imaging in this paper. Although some good performance in recovering or denoising LDPET images were reported, those deep learning based methods mentioned above were not evaluated quantitatively for lesion SUVs, which limited their usage in clinical applications. In order to

evaluate the clinical performance of our model, we also proposed a simulation framework to produce a series of simulation data to mimic complex clinical situations. The S-CycleGAN model was then applied to the clinical and simulated LDPET datasets (10% and 30% of FDPET datasets), and studied qualitatively and quantitatively.

## Methods

The goal of this work is to train a model to learn the non-linear mapping between LDPET and FDPET images. As shown in Fig 1, the proposed network is based on a CycleGAN architecture.

The proposed model includes two generators and two discriminators. We denote $G_{AB}$ is the mapping from LDPET domain (A) to FDPET domain (B), the $G_{BA}$ represents the opposite direction mapping. In addition, there are two discriminators $D_A$ and $D_B$ which intend to identify whether the output of each generator is real or fake. Then, we train the generators and discriminators simultaneously. Thus, we have the following optimization problem:

$$\min_{G_{AB},G_{BA}} \max_{D_A,D_B} \mathcal{L}(G_{AB}, G_{BA}, D_A, D_B) \tag{1}$$

Our proposed network combines four types of loss functions: adversarial loss ($\mathcal{L}_{adv}$), cycle-consistency loss ($\mathcal{L}_{cyclic}$), identity loss ($\mathcal{L}_{identity}$) and supervised learning loss ($\mathcal{L}_{sup}$). Therefore, the overall loss is defined by:

$$\mathcal{L} = \mathcal{L}_{adv} + \alpha\mathcal{L}_{cyclic} + \beta\mathcal{L}_{identity} + \gamma\mathcal{L}_{sup} \tag{2}$$

where $\alpha$, $\beta$ and $\gamma$ are hyperparameters.

**Adversarial loss**: We employ adversarial losses to generate image samples to obey the empirical distributions in the source and target domains. To improve the training stability of GANs, we apply the 1-Wasserstein distance [37] instead of the original log-likelihood function. The 1-Wasserstein distance or Earth-Mover (EM) distance is defined as follows.

$$W(P_r, P_g) = \inf_{\gamma \in \Pi(P_r, P_g)} E_{(x,y)\sim\gamma}[\|x - y\|] \tag{3}$$

Where $\Pi(P_r, P_g)$ denotes the set of all joint distributions $\gamma(x, y)$ whose marginals are respectively $P_r$ and $P_g$.

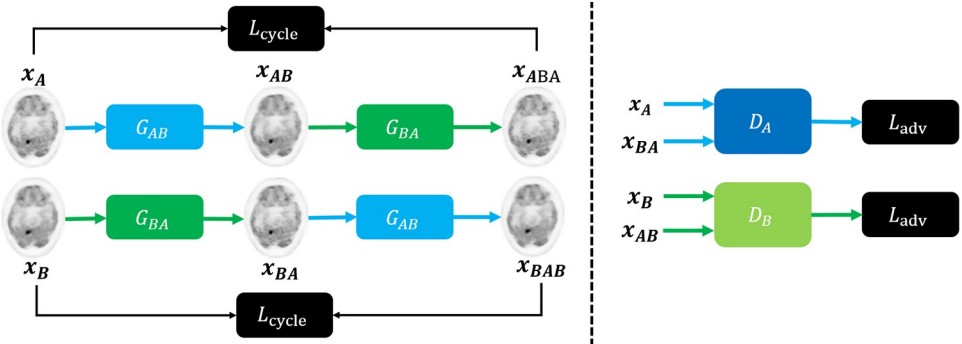

**Fig 1. S-CycleGAN framework.** Overview of the proposed framework for LDPET image recovery.

Thus, the adversarial objective function $\mathcal{L}(G_{AB}, D_B)$ is defined as follows.

$$\min_{G_{AB}} \max_{D_B} \mathcal{L}(G_{AB}, D_B) = -E_{x_B \sim P_B}[D_B(x_B)]$$

$$+E_{x_A \sim P_A}[D_B(G_{AB}(x_A))] + \lambda E_{\tilde{y}}[(\|\nabla_{\tilde{y}} D_A(\tilde{y})\|_2 - 1)^2] \tag{4}$$

Where $\lambda$ is a regularization parameter, which controls the trade-off between the Wasserstein distance and the gradient penalty term. $\tilde{y}$ is uniformly sampled along straight lines for pairs of $G_{AB}(x_A)$ and $x_B$. The adversarial loss for the reverse direction $\mathcal{L}(G_{BA}, D_A)$ is defined in a similar way. The final *adversarial loss*($\mathcal{L}_{adv}$) is defined

$$\mathcal{L}_{adv} = \frac{1}{2}(\mathcal{L}(G_{AB}, D_B) + \mathcal{L}(G_{BA}, D_A)) \tag{5}$$

**Cycle consistency loss**: We adopt a cycle consistency term that the FDPET and LDPET images could be transformed mutually as an additional regularization to help learning of $G_{AB}$ and $G_{BA}$. The cyclic loss is defined by

$$\mathcal{L}_{cyclic}(G_{AB}, G_{BA}) = E_{x_A \sim P_A}[\|G_{BA}(G_{AB}(x_A)) - x_A\|_1]$$

$$+E_{x_B \sim P_B}[\|G_{AB}(G_{BA}(x_B)) - x_B\|_1] \tag{6}$$

Where $\|:\|_1$ denotes the $l_1$-norm. This allows for additional information to be shared between LDPET and FDPET images in learning their corresponding generators.

**Identity loss**: In real clinical situation, the input to the generator $G_{AB}$ can be a full-dose image, but we expect the generator does not alter such clean image, and vice versa. Besides, indentity loss provides another regularization in the training procedure and is formulated as follows:

$$\mathcal{L}_{identity}(G_{AB}, G_{BA}) = E_{x_A \sim P_A}[\|G_{BA}(x_A) - x_A\|_1]$$

$$+E_{x_B \sim P_B}[\|G_{AB}(x_B) - x_B\|_1] \tag{7}$$

**Supervised learning loss**: Since we have paired datasets, we can train our model in a supervised fashion. Then, we can define a supervision loss as follows:

$$\mathcal{L}_{sup}(G_{AB}, G_{BA}) = E_{x_A \sim P_A}[\|G_{AB}(x_A) - x_B\|_1]$$

$$+E_{x_B \sim P_B}[\|G_{BA}(x_B) - x_A\|_1] \tag{8}$$

## Network architecture

Our proposed model, S-CycleGAN, is constituted of two generator networks, $G_{AB}$ and $G_{BA}$, and two discriminator networks, $D_A$ and $D_B$. The generator networks take one domain's image and estimate another domain's image. The discriminator networks aim to differentiate between the real and estimated image.

**Generative networks**: The network architecture of two generators $G_{AB}$ and $G_{BA}$ is illustrated in Fig 2. The basic structure is optimized for the LDCT image denoising in [38, 39]. To reduce network complexity and adapt to PET image, we set the filter number to 64 instead of 128 in the original model and add a ReLU layer before model output. As shown in Fig 2, the first two convolution layers use 64×3×3 convolution kernels to produce 64 feature maps, and connect to 6 sets of residual modules, where each module is composed of 3 sets of convolution, batch normalization, and a ReLU layer, and one residual connection with a ReLU layer. Later

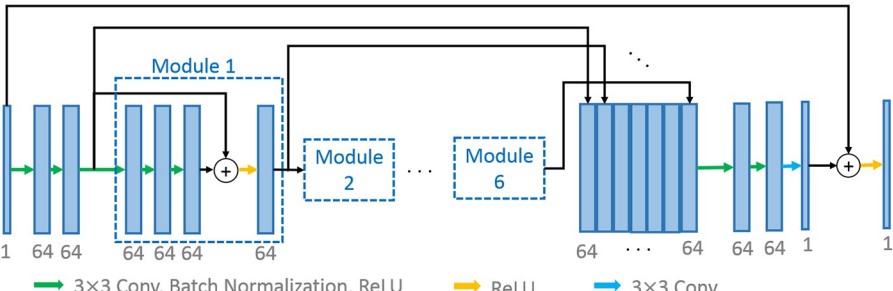

**Fig 2. Generator.** A generator architecture in the proposed framework.

on, a concatenation layer that concatenates the inputs of each module and the output of the last module, and two convolution layer with 64 feature maps are applied. Finally, the last convolution layer with 3×3 convolution kernel combined with an end-to-end bypass connection and an additional ReLU layer are used to estimate the FDPET image.

**Discriminator**: The discriminators take either a real PET image or an estimated one as input, and determines whether the input is real or not. As shown in Fig 3, the discriminator network is designed to have 4 stages of convolutions, followed by two fully-connected layers, of which the first has 1024 outputs and the last has 1 output. We apply 4×4 filter size for the convolution layers which have different numbers of filters as 64, 128, 256, 512 respectively. In addition, we use Leaky ReLU activation in the discriminator for all layers, with slope 0.2.

## Experimental setup

### Datasets

We trained our model by using human brain datasets. A total of 109 clinic patient (range 44.3-103kg) PET/CT images were taken by the Minfound ScintCare PET/CT 720L scanner with injection of 370.81±64.38 MBq of 18F-fluorodeoxyglucose(FDG), and we randomly selected 89, 10 and 10 patient data for training, validation and testing, respectively. All scans were taken about 5 minutes and usually start at 45-60 minutes later after injection. The reconstruction was performed using the manufacturer-provided software with all physical corrections, including attenuation, scatter, randoms, dead-time and SUV correction. The size of each 3D reconstructed PET image is 192×192×96 with pixel size of 2.1 mm. FDPET and LDPET images were reconstructed with the same parameters and post filters to ensure comparable spatial

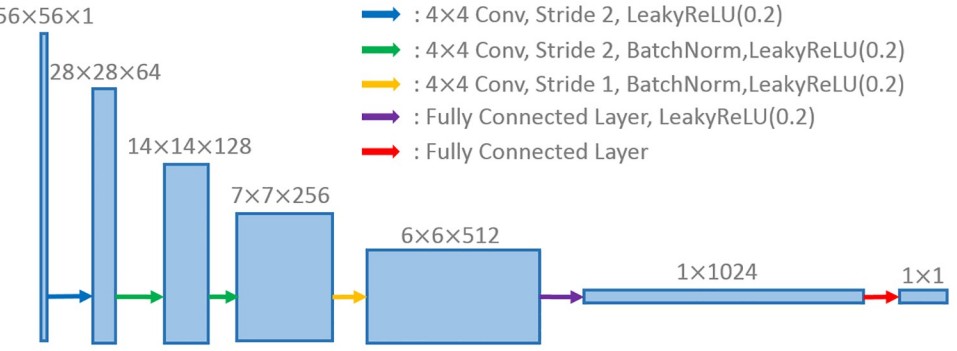

**Fig 3. Discriminator.** A discriminator architecture in the proposed framework.

resolution in both images. Two different simulated doses, i.e. 10% and 30% counts of original scan, where generated by randomly discarding events in FDPET list mode data. Although the 10% and 30% images were generated by emulated low-count scans, however, those images have comparable quality with actual low-dose scan, and confirmed by the recent work [40]. By this way, FDPET and LDPET images are spatially aligned.

In order to evaluate the clinical feasibility of our proposed model, a Monte Carlo simulation framework using GATE [41, 42] was carefully designed and shown in Fig 4. At the first step, lesion maps with different shapes, sizes, and locations were extracted from a few of known patient's datasets (different from the above 109 patients), and a patient's attenuation map (μ-map) was generated from its corresponding CT image. Then, these two maps were fed into GATE, and were simulated with the same system settings as Minfound ScientCare PET/CT 720L scanner. Finally, the simulated coincidence data of lesions combined with the clinical coincidence data of the patient were reconstructed by the manufacturer-provided software to produce the final PET image. To systematically evaluate model performance, a series of simulation data with various activities, sizes and shapes were produced and reconstructed. In order to reduce statistical variations, each simulation configuration was repeated 3 times and total 45 simulations were used in later quantitative evaluations. The details of those lesions are also provided in Table 1 and Fig 5. The activity for background is about 30 kBq/ml. Some typical simulated PET images are shown in Fig 6.

In order to reduce the computational cost for training, we extracted overlapping patches from LDPET and FDPET images instead of directly feeding entire PET images to the training pipeline. We cropped LDPET and FDPET images into patches of 56×56 at the same place for the supervised learning with sliding step 40. In total, there are 136704 and 15360 patches for training and validation. Since PET images have a large range in pixel values, we scaled pixel values to [0, 1].

## Evaluation measures

Six measures are used to evaluate the model performance including the normalized root mean square error (NRMSE), structural similarity index (SSIM [43]), peak signal-to-noise ratio (PSNR), learned perceptual image patch similarity (LPIPS [44]), relative errors (RE) for $SUV_{mean}$ and $SUV_{max}$, which are defined as following.

$$SSIM = \frac{2\mu_x\mu_y + C_1}{\mu_x^2 + \mu_y^2 + C_1} * \frac{2\sigma_{xy} + C_2}{\sigma_x^2 + \sigma_y^2 + C_2} \tag{9}$$

$$= l(i,j) * cs(i,j)$$

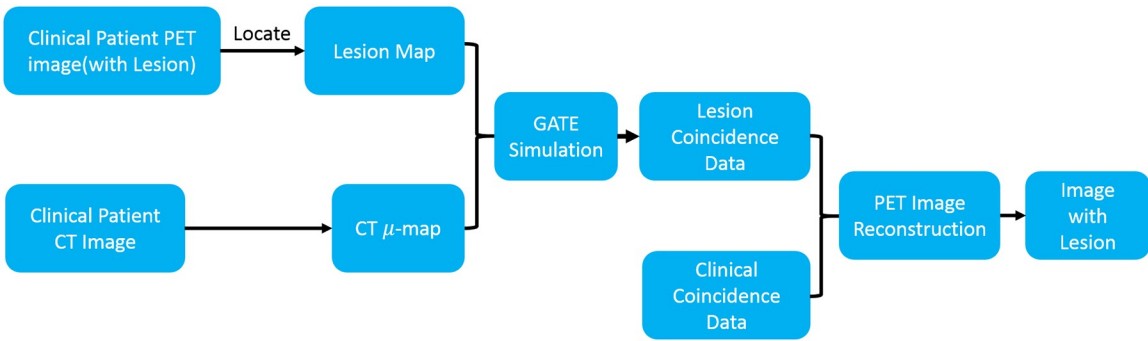

**Fig 4. Simulation architecture.** A GATE based simulation architecture.

**Table 1. Lesion position and size information.** Where $D_{max}$ and $D_{mean}$ are the maximum and mean diameters, respectively.

| No. | Center Position (x,y,z) | $D_{max}$ [$mm$] | $D_{mean}$ [$mm$] | Volume [$mm^3$] | Label |
|---|---|---|---|---|---|
| 1 | 99,92,29 | 9.39 | 7.72 | 240.78 | small |
| | | 16 | 9.72 | 481.57 | middle |
| | | 18.78 | 12.25 | 963.14 | large |
| 2 | 108,62,49 | 8.9 | 7.41 | 213.0 | small |
| | | 12.2 | 9.27 | 416.74 | middle |
| | | 16 | 11.67 | 833.49 | large |
| 3 | 95,86,61 | 17.3 | 9.91 | 509.36 | small |
| | | 18.78 | 12.45 | 1009.45 | middle |
| | | 21.9 | 15.68 | 2018.9 | large |
| 4 | 74,68,43 | 10.5 | 7.5 | 222.26 | small |
| | | 13.5 | 9.05 | 388.96 | middle |
| | | 16.2 | 11.32 | 759.4 | large |
| 5 | 98,35,57 | 12.4 | 8.09 | 277.83 | small |
| | | 16.4 | 9.85 | 500.09 | middle |
| | | 16.8 | 12.33 | 981.67 | large |

$$NRMSE = \sqrt{\frac{\sum_{i=1}^{N} \sum_{j=1}^{M} (x_{ij} - y_{ij})^2}{\sum_{i=1}^{N} \sum_{j=1}^{M} y_{ij}^2}} \times 100 \tag{10}$$

$$PSNR = 20 * log_{10}(\frac{MAX}{MSE}) \tag{11}$$

$$RE = (\frac{SUV_{Model} - SUV_{FDPET}}{SUV_{FDPET}}) \times 100 \tag{12}$$

Where $C_1$ and $C_2$ are constants, $\mu_x$, $\mu_y$, $\sigma_x$, $\sigma_y$, and $\sigma_{xy}$ are mean and standard deviation in the patch centered at pixel (i,j). MAX is the peak intensity of the image, MSE is absolute mean square error.

The SUV (standardized uptake value) is commonly used as a relative measure of FDG uptake. The basic expression for SUV [32] is

$$SUV = \frac{r}{a'/w} \tag{13}$$

Where r is the radioactivity activity concentration [kBq/ml] measured by the PET scanner within a region of interest (ROI), $a'$ is the decay-corrected amount of injected radiolabeled

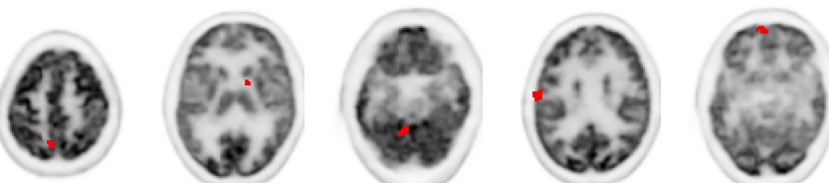

**Fig 5. Embedded lesions.** The distribution of 5 embedded lesions in the GATE simulation.

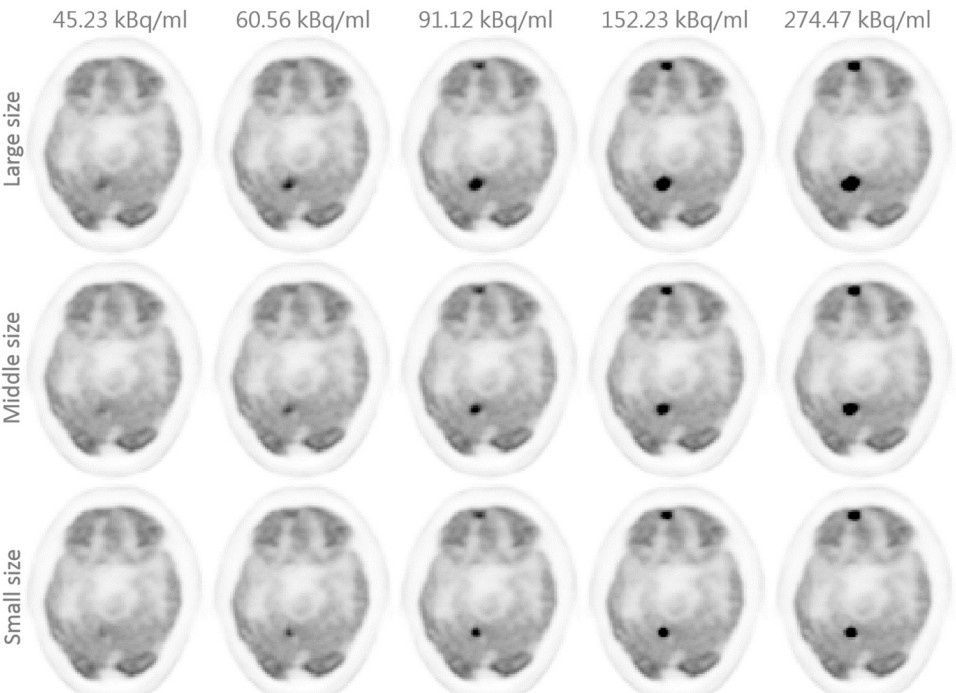

**Fig 6. Simulation PET images.** Typical FDPET images with embedded lesions using the proposed simulation architecture.

FDG (kBq), and w is the weight of the patient (g), which is used as a surrogate for a distribution volume of tracer.

## Implementation details

In the proposed model, training was performed by minimizing the loss function 2. We utilized the Adam optimizer [45] with $\beta_1 = 0.5$ and $\beta_2 = 0.999$ to minimize the total loss function of the proposed network. We set the learning rate to $2\times10^{-4}$, hyperparameters $\alpha = 10$, $\beta = 5$, and $\gamma = 5$. The trade-off parameter $\lambda$ between Wasserstein distance and gradient penalty was set to be 10, as [37] suggested. The hyperparameters basically were derived from the original CycleGAN paper. As for the parameter, $\gamma$, it was determined by experiment in order to get a trade-off between the noise and the bias of SUV at the lesion regions. The size of the patch was set to 56×56 and the mini-batch is 16. Kernels were initialized randomly from a Gaussian distribution. All experiments were conducted using Keras [46] with Tensorflow backend on a NVIDA TITAN GTX GPU.

The training epoch was set to 200 based on experience with early-stop strategy when the validation loss is minimal (the patience value is 5). It takes 7 days for training at current GPU hardware. Although the training was done on patches, the proposed network can process images of arbitrary sizes. All the testing images were simply fed into the network without decomposition and required 74ms of inference time per image slice.

## Experimental results

### Comparison with other methods

To study the effectiveness of our proposed model, we compared it with RED-CNN [15] and 3D-cGAN [21]. The network structure and parameters of these competing methods were set

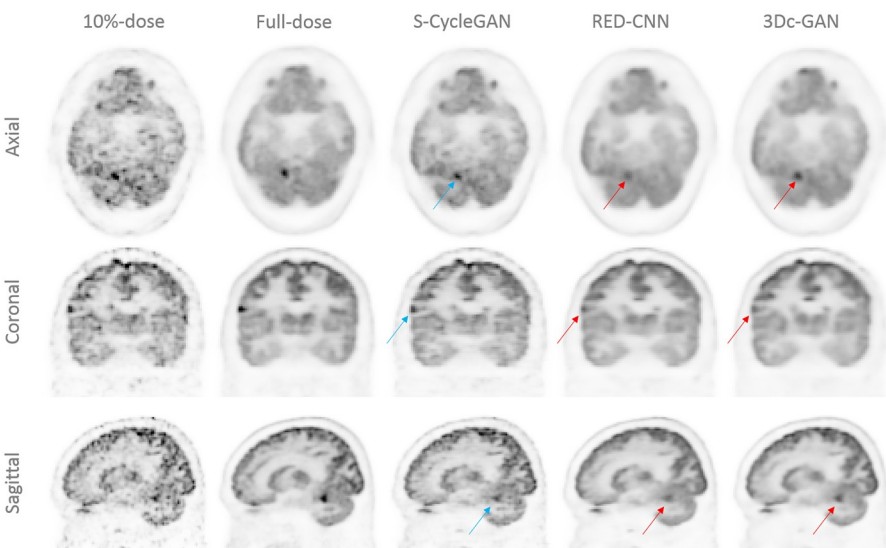

**Fig 7. Model comparison for 10% of FDPET.** Qualitative comparison of PET images by S-CycleGAN, RED-CNN and 3D-cGAN, where low-dose data is 10% of FDPET.

per the suggestions from the original papers and re-implemented by Keras. For a qualitative comparison, some sample images of the predicted FDPET from three deep learning methods, the corresponding LDPET and FDPET reconstruction are shown in Figs 7 and 8 for 10% and 30% dose of FDPET, respectively. The estimated images by all deep learning methods show better image quality than low-dose images, providing better noise reduction and structure details recovery.

The quantitative measures in terms of NRMSE, SSIM and PSNR are shown in Table 2 using 10 testing patient datasets. All three predicted images have better noise control and structure similarities than low-dose images, but similar peak signal to noise ratios. RED-CNN and 3D-

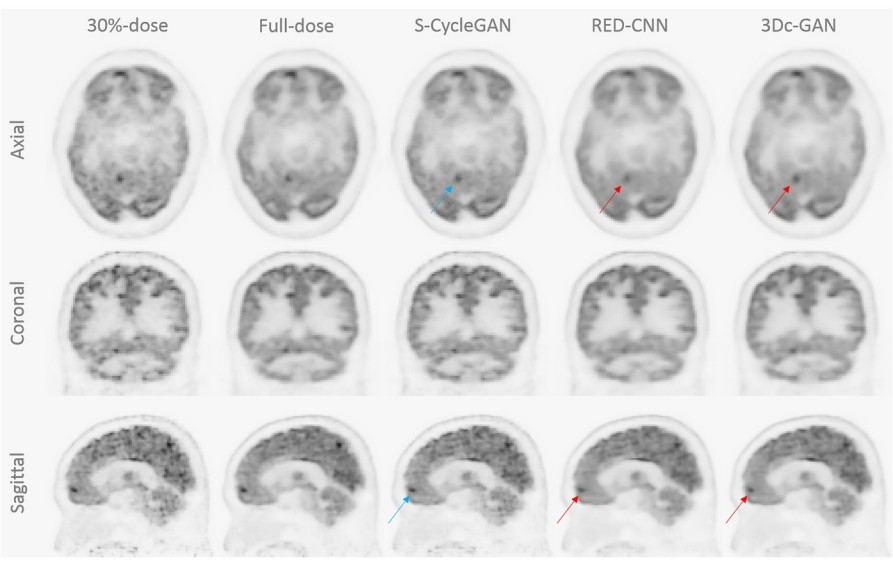

**Fig 8. Model comparison for 30% of FDPET.** Qualitative comparison of PET images by S-CycleGAN, RED-CNN and 3D-cGAN, where low-dose data is 30% of FDPET.

**Table 2. Quantitative comparison on normal subjects (all 10 patients).**

| Method | Dose Level | NRMSE Mean±std. | SSIM Mean±std. | PSNR(dB) Mean±std. |
|---|---|---|---|---|
| Low-dose | 10% | 25.985±5.962 | 0.962±0.0186 | **66.408±1.654** |
| | 30% | 12.218±2.567 | 0.991±0.00388 | 67.208±1.954 |
| S-CycleGAN | 10% | 17.531±3.702 | 0.981±0.00803 | 65.861±1.918 |
| | 30% | 10.405±2.056 | 0.994±0.00262 | **67.185±2.176** |
| RED-CNN | 10% | **13.838±2.606** | 0.989±0.00510 | 64.886±1.780 |
| | 30% | **8.901±1.781** | 0.994±0.00212 | 66.901±1.995 |
| 3Dc-GAN | 10% | 14.021±2.296 | **0.989±0.00415** | 64.964±1.998 |
| | 30% | 9.395±1.682 | **0.995±0.00203** | 67.034±2.019 |

cGAN models have better NRMSE scores than S-CycleGAN, however, their predicted images suffer from over-smoothing issues and may compromise the diagnostic performance, as shown in Figs 8 and 7 (indicated by red arrows).

As suggested by Zhang et al. [47], the traditional metrics (L2/PSNR, SSIM, FSIM) disagree with human judgments, a learned perceptual image patch similarity metric was proposed to evaluate image quality. The LPIPS measurements between model prediction and FDPET is shown in Fig 9. The estimated images by all deep learning methods show better LPIPS scores than low-dose images and S-CycleGAN obtains the best score. The average LPIPS scores of LDPET (30% of FDPET), S-CycleGAN, RED-CNN and 3D-cGAN are 0.035, 0.026, 0.031 and 0.031, respectively.

## Clinical evaluation for specific VOIs

In clinic, the mean and maximum SUVs are often used as bases for diagnosis to characterize suspicious high uptakes [31, 32]. Therefore, the SUV measures are used to investigate the

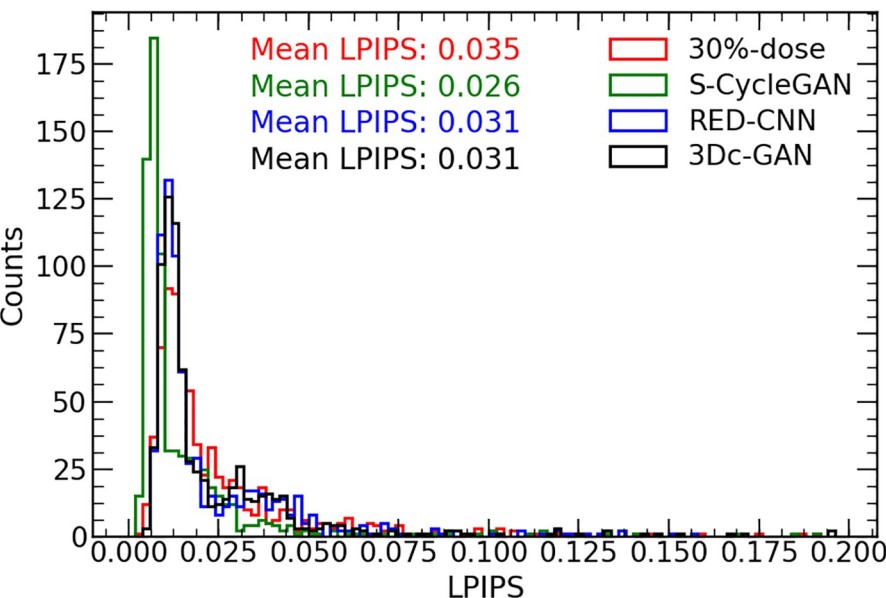

**Fig 9. LPIPS score comparison for 30% of FDPET.** LPIPS score (smaller is better) comparison of PET images by LDPET (30% of FDPET), S-CycleGAN, RED-CNN and 3D-cGAN.

**Table 3. The average bias and standard deviation for $SUV_{mean}$ of lesion tissues recovering from 10% of FDPET datasets.**

| Lesion size | Method | 45.23 [kBq/ml] | 60.56 [kBq/ml] | 91.12 [kBq/ml] | 152.23 [kBq/ml] | 274.47 [kBq/ml] |
|---|---|---|---|---|---|---|
| Small | S-CycleGAN | **-7.38±4.65** | **-12.03±5.98** | **-11.00±4.82** | **-7.18±4.80** | **-5.17±5.21** |
| | RED-CNN | -15.48±3.74 | -24.88±8.03 | -31.67±11.29 | -21.70±13.93 | -24.31±16.54 |
| | 3Dc-GAN | -13.49±6.48 | -21.46±9.58 | -22.30±10.09 | -21.45±10.44 | -34.78±10.41 |
| Middle | S-CycleGAN | **-6.90±4.61** | **-10.44±3.93** | **-7.12±5.15** | **-4.20±3.39** | **-2.97±3.38** |
| | RED-CNN | -12.29±5.61 | -21.45±8.70 | -25.08±10.41 | -13.92±12.28 | -16.57±14.28 |
| | 3Dc-GAN | -11.90±6.97 | -21.16±7.90 | -18.55±9.01 | -18.15±13.15 | -26.62±9.52 |
| Large | S-CycleGAN | **-6.92±2.13** | **-9.75±4.85** | **-4.56±3.32** | **-1.82±2.50** | **-0.81±1.22** |
| | RED-CNN | -12.43±5.25 | -20.28±8.19 | -19.12±10.10 | -11.77±12.17 | -14.39±11.26 |
| | 3Dc-GAN | -13.42±4.74 | -21.33±7.22 | -18.14±10.23 | -18.42±10.13 | -21.19±10.02 |

effectiveness of the proposed method for specific VOIs on both the normal and lesion tissues. The datasets were produced by our proposed simulation framework as mentioned in Experimental Setup section. In this analysis, the mean and maximum SUV biases and deviations were evaluated for all deep learning models mentioned above.

The average biases and standard deviations of $SUV_{mean}$ and $SUV_{max}$ of lesion tissues are shown in Tables 3, 4 and Tables 5, 6, respectively. Since $SUV_{max}$ is not critical for normal tissues, only $SUV_{mean}$ error is shown in Table 7. The results of different lesion sizes and FDG concentrations are also shown in above tables.

**Table 4. The average bias and standard deviation for $SUV_{mean}$ of lesion tissues recovering from 30% of FDPET datasets.**

| Lesion size | Method | 45.23 [kBq/ml] | 60.56 [kBq/ml] | 91.12 [kBq/ml] | 152.23 [kBq/ml] | 274.47 [kBq/ml] |
|---|---|---|---|---|---|---|
| Small | S-CycleGAN | **-0.55±5.66** | **-3.16±7.15** | **-5.14±4.35** | **-2.81±3.83** | **-2.62±2.48** |
| | RED-CNN | -6.59±6.27 | -11.55±8.39 | -11.09±5.38 | -3.08±4.74 | -4.76±8.27 |
| | 3Dc-GAN | -5.27±5.84 | -8.67±6.47 | -8.87±3.17 | -11.41±3.60 | -16.68±6.51 |
| Middle | S-CycleGAN | **-1.50±4.84** | **-3.73±3.52** | **-2.99±1.65** | **-1.37±1.31** | **-1.17±1.95** |
| | RED-CNN | -5.95±6.05 | -10.81±5.64 | -7.96±3.24 | -1.88±4.90 | -3.33±6.95 |
| | 3Dc-GAN | -6.05±5.53 | -9.27±4.88 | -7.02±3.05 | -11.61±5.15 | -14.67±3.07 |
| Large | S-CycleGAN | **-1.90±1.79** | **-3.89±1.03** | **-2.13±1.59** | **-0.38±1.07** | **-0.35±1.09** |
| | RED-CNN | -6.29±3.58 | -9.67±4.56 | -6.27±4.09 | -2.31±4.15 | -3.60±4.73 |
| | 3Dc-GAN | -6.74±4.09 | -9.36±3.82 | -6.49±4.95 | -7.72±5.03 | -10.16±4.19 |

**Table 5. The average bias and standard deviation for $SUV_{max}$ of lesion tissues recovering from 10% of FDPET datasets.**

| Lesion size | Method | 45.23 [kBq/ml] | 60.56 [kBq/ml] | 91.12 [kBq/ml] | 152.23 [kBq/ml] | 274.47 [kBq/ml] |
|---|---|---|---|---|---|---|
| Small | S-CycleGAN | **-2.52±18.50** | **-12.75±13.81** | **-6.86±8.65** | **-4.95±8.08** | **-5.01±8.25** |
| | RED-CNN | -22.14±10.95 | -38.52±12.44 | -44.96±13.78 | -22.99±20.91 | -24.25±25.41 |
| | 3Dc-GAN | -22.53±10.56 | -37.25±11.82 | -32.85±14.51 | -24.20±17.38 | -40.52±16.64 |
| Middle | S-CycleGAN | **-1.53±18.13** | **-10.22±20.20** | **-0.03±23.77** | **-9.99±10.11** | **-3.26±7.59** |
| | RED-CNN | -19.76±11.16 | -38.95±8.91 | -38.01±15.41 | -18.49±14.93 | -12.36±20.97 |
| | 3Dc-GAN | -19.28±12.50 | -38.26±10.47 | -28.79±17.26 | -22.40±16.16 | -31.75±15.05 |
| Large | S-CycleGAN | **-1.42±17.67** | **-7.52±23.74** | **4.95±19.97** | **4.05±12.46** | **1.28±4.22** |
| | RED-CNN | -23.52±7.92 | -35.11±12.13 | -22.94±18.51 | -6.85±16.13 | -2.42±18.15 |
| | 3Dc-GAN | -22.36±8.82 | -37.36±14.17 | -25.15±15.89 | -17.86±15.92 | -18.41±19.70 |

**Table 6. The average bias and standard deviation for $SUV_{max}$ of lesion tissues recovering from 30% of FDPET datasets.**

| Lesion size | Method | 45.23 [kBq/ml] | 60.56 [kBq/ml] | 91.12 [kBq/ml] | 152.23 [kBq/ml] | 274.47 [kBq/ml] |
|---|---|---|---|---|---|---|
| Small | S-CycleGAN | **-1.44±8.87** | **-6.49±9.43** | **-10.58±4.59** | -6.52±6.45 | **-4.91±3.92** |
| | RED-CNN | -15.36±10.89 | -24.32±8.96 | -19.56±6.54 | **-6.09±6.10** | -5.37±14.29 |
| | 3Dc-GAN | -11.76±7.59 | -18.75±6.06 | -14.00±7.20 | -16.96±5.70 | -19.79±9.37 |
| Middle | S-CycleGAN | **-4.05±7.01** | **-8.32±2.41** | **-6.49±3.70** | -4.50±3.67 | -2.56±5.20 |
| | RED-CNN | -15.59±8.45 | -23.36±5.64 | -14.97±5.90 | **-2.59±5.63** | **0.91±11.38** |
| | 3Dc-GAN | -12.75±7.15 | -20.06±4.77 | -11.37±7.51 | -18.67±5.90 | -16.89±9.29 |
| Large | S-CycleGAN | **-2.75±3.31** | **-6.96±4.73** | **-3.89±5.56** | **-0.14±3.88** | **0.54±4.78** |
| | RED-CNN | -16.73±5.01 | -20.56±6.84 | -10.52±7.31 | 0.34±6.44 | 1.32±7.78 |
| | 3Dc-GAN | -13.16±6.03 | -17.79±7.46 | -5.50±11.45 | -5.98±8.82 | -5.91±11.22 |

**Table 7. The bias and deviations for $SUV_{mean}$ of normal tissues.**

| Dose Level | S-CycleGAN Mean±std. | RED-CNN Mean±std. | 3Dc-GAN Mean±std. |
|---|---|---|---|
| 10% | **-3.11±7.14** | -4.42±7.60 | -3.49±7.77 |
| 30% | -0.78±3.08 | -1.15±3.16 | **-0.57±3.72** |

$SUV_{mean}$ **deviation**: As observed in Table 7, all the models have very similar SUV-mean values in normal tissues which biases are less than 5% for both 10% and 30% dose levels. However, as seen in Tables 3 and 4, the RED-CNN and 3D-cGAN have much larger biases than S-CycleGAN in lesion tissues, especially for smaller lesion sizes and lower activities. The average $SUV_{mean}$ biases of S-CycleGAN, RED-CNN and 3D-cGAN for all lesions and activities are -6.4±5.3%, -18.7±11.8% and -20.0±10.8% for 10% dose level, and -2.8±4.1%, -6.3±6.4% and -9.8±6.0% for 30% dose level, respectively. It can be also seen that the biases and deviations of $SUV_{mean}$ decrease as the lesion size and activities increases for S-CycleGAN model in most cases. Those observations indicate the good robustness of our proposed model.

$SUV_{max}$ **deviation**: The $SUV_{max}$ results of all three deep learning methods are shown in Tables 5 and 6 for 10% and 30% dose of FDPET, respectively. Since the single pixel value in the VOI is largely affected by the statistical property of data, the $SUV_{max}$ values in LDPET images have large biases and deviations, especially for the lower dose level. Our proposed S-CycleGAN model is trending to reduce the biases and deviations but this ability gets worse as lesion sizes decrease. The average $SUV_{max}$ biases of S-CycleGAN, RED-CNN and 3D-cGAN for all lesions and activities are -3.7±16.2%, -24.9±11.7% and -28.0±16.5% for 10% dose level, and -5.2±6.8%, -11.4±11.7% and -14.8±9.8% for 30% dose level, respectively. Those results are suggesting the S-CycleGAN method can better preserve the $SUV_{max}$ values than other two methods.

## Ablation study

**Impact of supervised learning loss**: The impact of the supervised learning loss was studied for the proposed model. A modified model, named as CycleGAN, was trained and tested with all the loss functions except the supervised loss $L_{sup}$. Image artifacts of missing structures are observed in about 7% of the slices generated by the CycleGAN model, as indicated by the red and yellow rectangle in Fig 10. Therefore, the use of supervised learning loss could reduce these artifacts and maintaining the fidelity of the PET image.

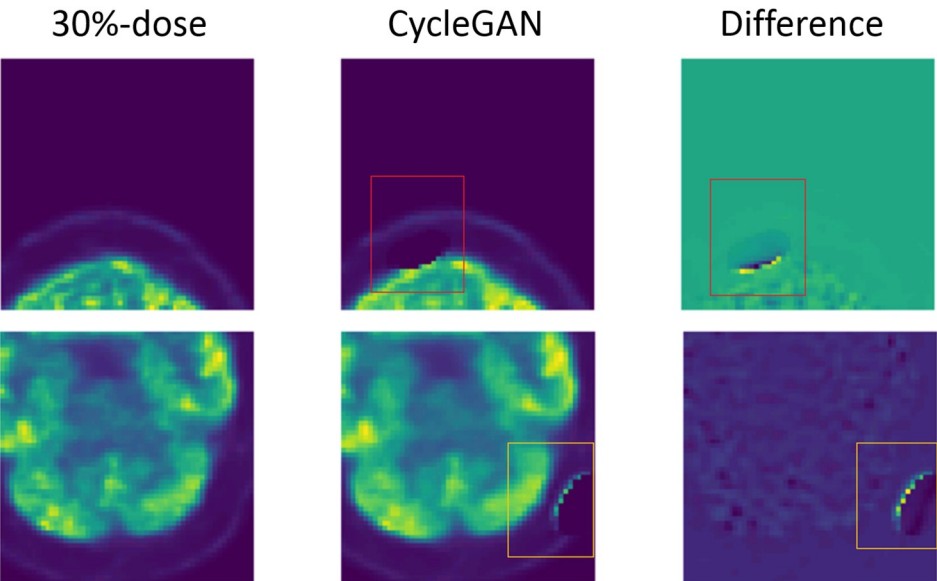

**Fig 10. LDPET denoising with CycleGAN.** A typical PET slice image with artifact from CycleGAN.

**Impact of cycle-consistency loss**: The effectiveness of cycle consistence loss was also studied by comparing the S-CycleGAN and 3D-cGAN model which didn't involve this loss. As shown in Tables 3, 4, 5 and 6, S-CycleGAN model can better preserve the $SUV_{mean}$ and $SUV_{max}$ values than 3D-cGAN, which indicates the effectiveness and necessity of the cycle consistence loss even if it is originally designed for the unpaired datasets training.

## Discussion

In order to systematically evaluate model performance, we designed a novel simulation framework to produce clinical-like data, in which the embedded lesions are exacted from the clinical data by considering realistic structures, sizes, activities and dose levels. Such simulations would be helpful to understand the clinical performance of the proposed method since it is almost impossible to know the true lesion uptakes in clinic. Moreover, this method can be extend to other related model performance studies.

Although our model has achieved the compelling results, there still exist some limitations. Our proposed model, S-CycleGAN, requires longer training time than other standard GAN-based and CNN-based methods. The future work should consider more efficient architectures. Though this paper mainly focuses on PET brain images, the same model with different hyper-parameters has been applied to PET body images too. More results will be presented in the near future once enough PET body datasets are acquired and trained.

Recent published papers [20–22] on LDPET image recovery were extended to even lower doses. However, it is difficult to conclude which approach can reduce more dose since different paper uses different datasets, acquisition protocols and scanners. In this paper, the training set of 99 patients has 110±23M average coincidence counts. Consequently, our proposed S-CycleGAN model actually takes the count variation into account in the training and can be used for a relatively widespread dose levels in complicated clinical situations. The recently published paper [48] uses a very similar method, the CycleGAN, for LDPET denoising, but they still didn't investigate the metrics of $SUV_{max}$ and robustness to different count levels.

All of those approaches usually compared the structure similarity, noise, signal-to-noise ratio or $SUV_{mean}$, but none of them involved the evaluation on $SUV_{max}$. $SUV_{max}$ is more often used in the clinical practice due to its better reproducibility than $SUV_{mean}$ since the maximum value within a VOI (or region-of interest) is invariant with respect to small spatial shifts [32, 33]. Due to the supervised training mode, the $SUV_{mean}$ can be easily preserved, but not the $SUV_{max}$. From systematic study of $SUV_{mean}$ and $SUV_{max}$, our proposed model has demonstrated promising results in recovering a high quality image from a LDPET image. However, smaller lesion size and lower activity actually degrade the performance of all models compared in this paper.

As shown in Tables 3, 4, 5 and 6, the $SUV_{mean}$ values can be relatively easier to be preserved as compared the $SUV_{max}$ values. Our proposed model has demonstrated better quantitative results than RED-CNN and 3D-cGAN no matter which dose level is used. When predicted images in the same dose level are compared, the $SUV_{mean}$ values show strong dependence on lesion sizes and activity concentrations. On the other hand, the $SUV_{max}$ values only show strong dependence on lesion sizes, but not the activity concentrations. Moreover, the $SUV_{max}$ values still have quite large variations even though 45 simulations are used in evaluations. These phenomena can be partially explained by two factors: one is image noise caused by data itself and reconstruction/post-processing methods, which can strongly affect $SUV_{max}$ values relying only on the single pixel values; another is the partial volume effect caused by the finite system spatial resolution and image sampling, which can heavily reduce the accuracy of $SUV_{mean}$ and $SUV_{max}$ values, especially for the smaller volumes of VOIs, or the lower activity ratio between VOIs and their surrounding background [49].

As compared to the 30% dose level, the images recovered from the 10% dose level still have good scores in normal tissues in terms of NRMSE, SSIM and PSNR, but much larger biases and deviations for the $SUV_{mean}$ and $SUV_{max}$ in lesion tissues for all three deep learning methods. This alerts us a potential risk that any diagnosis relying on these two indexes could be changed in clinical practice. Therefore, we should be cautious in developing any deep learning approaches which could largely change $SUV_{mean}$ and $SUV_{max}$ while reducing dose. For this reason, the 30% dose level is preferred in this study since it can better balance the tradeoff between SUV values and dose reduction.

## Conclusion

In conclusion, we have introduced a novel deep learning based generative adversarial model with the cycle consistent to estimate the high-quality image from the LDPET image. The proposed S-CycleGAN approach has produced comparable image quality as corresponding FDPET images by suppressing image noise and preserving structure details in a supervised learning fashion. Systemic evaluations further confirm that the S-CycleGAN approach can better preserve mean and maximum SUV values than other two deep learning methods, and suggests the amount of dose reduction should be carefully decided according to the acquisition protocols and clinical usages.

## Acknowledgments

The authors would like to thank many colleagues at Zhejiang University and MinFound Medical Systems Co., Ltd. for the helpful discussion and PET image reconstruction.

## Author Contributions

**Conceptualization:** Kui Zhao.

**Data curation:** Kui Zhao, Size Gao, Yaofa Wang, Xin Zhao, Huatao Wang, Kanfeng Liu, Yunqi Zhu.

**Formal analysis:** Xiaozhuang Wang.

**Investigation:** Long Zhou.

**Methodology:** Kui Zhao, Size Gao, Xin Zhao.

**Project administration:** Hongwei Ye.

**Resources:** Yaofa Wang, Xin Zhao, Huatao Wang, Yunqi Zhu.

**Software:** Long Zhou, Size Gao, Xiaozhuang Wang, Yaofa Wang, Kanfeng Liu, Yunqi Zhu.

**Supervision:** Kui Zhao.

**Validation:** Long Zhou.

**Visualization:** Long Zhou.

**Writing – original draft:** Long Zhou.

**Writing – review & editing:** Kui Zhao, Size Gao, Yaofa Wang, Xin Zhao, Huatao Wang, Kanfeng Liu, Yunqi Zhu, Hongwei Ye.

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
