## [Decision Letter · Decision Letter 0]

30 Apr 2020

PONE-D-19-34960

Study of low-dose PET image recovery using supervised learning with CycleGAN

PLOS ONE

Dear Dr. Ye,

Thank you for submitting your manuscript to PLOS ONE. After careful consideration, we feel that it has merit but does not fully meet PLOS ONE’s publication criteria as it currently stands. Therefore, we invite you to submit a revised version of the manuscript that addresses the points raised during the review process.

ACADEMIC EDITOR: 

The two reviewers are in agreement that the paper is well written and interesting, but they also raise several important concerns, including i) methodological details missing, ii) additional experiments to conduct in order to increase the confidence in the conclusions (currently not entirely supported by the results).

I also suggest the authors to evaluate the produced images using other, more advanced metrics than SUVmax / mean. For instance, radiomic features have been used increasingly over the last decade to characterize PET images further than basic SUV measurements.

We would appreciate receiving your revised manuscript by Jun 14 2020 11:59PM. To enhance the reproducibility of your results, we recommend that if applicable you deposit your laboratory protocols in protocols.io, where a protocol can be assigned its own identifier (DOI) such that it can be cited independently in the future. For instructions see: http://journals.plos.org/plosone/s/submission-guidelines#loc-laboratory-protocols

We look forward to receiving your revised manuscript.

Kind regards,

Mathieu Hatt, MSc, PhD, HDR

Academic Editor

PLOS ONE

Journal Requirements:

We note that one or more of the authors are employed by a commercial company: "Zhejiang Minfound Intelligent Healthcare Technology Co., Ltd. and MinFound Medical Systems Co., Ltd.,"

4. We note you have included a table to which you do not refer in the text of your manuscript. Please ensure that you refer to Table 2 in your text; if accepted, production will need this reference to link the reader to the Table.

Additional Editor Comments (if provided):

Reviewers' comments:

Reviewer's Responses to Questions

**Comments to the Author**

1. Is the manuscript technically sound, and do the data support the conclusions?

Reviewer #1: Partly

Reviewer #2: Partly

2. Has the statistical analysis been performed appropriately and rigorously? 

Reviewer #1: I Don't Know

Reviewer #2: No

3. Have the authors made all data underlying the findings in their manuscript fully available?

Reviewer #1: No

Reviewer #2: No

4. Is the manuscript presented in an intelligible fashion and written in standard English?

Reviewer #1: Yes

Reviewer #2: Yes

5. Review Comments to the Author

Reviewer #1: REVIEW: “Study of low-dose PET image recovery using supervised learning with CycleGAN”

Summary:

- In this paper, the authors tackle the problem of recovering FDPET images from LDPET images.

- They introduce a new architecture “S-CycleGAN” to learn a nonlinear mapping between LDPET and FDPET images. It is based on the well-known CycleGAN architecture but trained with an additional supervised learning loss.

- The system is compared with two other deep learning methods (RED-CNN and 3D-cGAN) with five different metrics including SUVmean and SUVmax.

Key strengths:

- The method is evaluated quantitatively for lesion SUVs with SUVmax and SUVmean. They show experimentally that it is better at preserving the SUVmax values than the two other methods.

- The paper is easy to read and understand.

Key weaknesses:

- The authors do not substantiate the choice of the CycleGAN architecture (designed for unpaired datasets) on a paired dataset. The benefit of adding a second generator/discriminator pair to transfer an image from the FDPET domain to the LDPET domain is unclear.

- This claim is unproven: “the proposed method is better at preserving image details than the other two methods”. The samples shown in figures 7 and 8 seem to support this claim, but a quantitative justification should be given with the proper metrics. Indeed, the NRMSE, SSIM, and PSNR are not adequate perceptual metrics (https://arxiv.org/abs/1801.03924). A perceptual loss (LPIPS) or a mean opinion score measure (MOS) could be used to support this statement.

Miscellaneous remarks:

- Typo in the abstract: “S-CylceGAN”

- Typo line 78: “supervise”

Reviewer #2: This paper investigates a supervised, cycle-consistent, conditional GAN to map low-dose PET brain images to full-dose images. Reported performance is promising compared to other deep learning frameworks. However, the study has two main limitations that need to be addressed before possible publication.

* A training set (99 samples) and a test set (10 samples) were defined. However, no validation set is mentioned, which is a severe limitation. In particular, it is not clear how the numerous hyperparameters (the alpha, beta and gamma loss weights, the learning rate, Adam optimizer's beta1 and beta2 parameters, the patch size, network hyperparameters, etc.) were set: if the number of hyperparameters is close to the number of test samples (10), then the selection criterion for these hyperparameters is very critical. Also, it is not clear how the stopping criterion was defined. Training is generally stopped when the validation loss is minimal. When is it stopped if no validation set was defined? When the test loss is minimal? Given the small number of samples (109), I recommend to follow a cross-validation strategy on the 99 development samples: the value of hyperparameters (including the number of training epochs) could be defined as those maximizing the cross-validation score. A final model would then be trained on the entire development set using the optimal hyperparameter values.

* Each of the four training losses (Adversarial Loss, Cycle Consistency Loss, Identity Loss, Supervised Learning Loss) seem relevant. However, it is not clear how each of them contribute to the overall performance. Note that the alpha, beta ang gamma weights are not reliable indicators in themselves. The added value of this paper would be much higher if the impact of each loss function was analyzed independently: this could be done by retraining the system without the loss function to analyze.

Minor comments are listed below:

* Other papers address the same task using CycleGANs (for instance "Yang Lei et al. Low dose PET imaging with CT-aided cycle-consistent adversarial networks, Proc. SPIE 11312, Medical Imaging 2020: Physics of Medical Imaging, 1131247 (16 March 2020); " ext-link-type="uri" xlink:type="simple">https://doi.org/10.1117/12.2549386"). Differences with these papers should be highlighted.

* The expression of the Wasserstein distance should be provided.

* Training details for RED-CNN and 3D-cGAN solution should also be provided. Moreover, methodological differences between these solutions and the proposed solution should be better highlighted.

* Why providing hardware details (NVIDA TITAN GTX GPU) if no computation times are given? Training times and inference times should be provided.

Typos:

* applicaions - applications

* an Machine Learning - a Machine Learning

6. PLOS authors have the option to publish the peer review history of their article (what does this mean?). If published, this will include your full peer review and any attached files.

Reviewer #1: No

Reviewer #2: No

---

## [Author Response · Author response to Decision Letter 0]

23 Jul 2020

We would like to thank editors and reviewers for their valuable time and very careful assessment of our manuscript. Following are our point-by-point answers to reviewers’ comments. Changes in the manuscript are highlighted by the red color. 

PONE-D-19-34960

Study of low-dose PET image recovery using supervised learning with CycleGAN

PLOS ONE

Reviewer #1: REVIEW: “Study of low-dose PET image recovery using supervised learning with CycleGAN”

Comments:

The authors do not substantiate the choice of the CycleGAN architecture (designed for unpaired datasets) on a paired dataset. The benefit of adding a second generator/discriminator pair to transfer an image from the FDPET domain to the LDPET domain is unclear.

As we know, the CycleGAN model is originally designed for unpaired data and style or feature transfer. However, in recent published studies (Harms et al. 2019), training images are paired by registration or as similar in structure as possible to preserve quantitative pixel values, lead network to focus on mapping details by removing large geometric mismatch, and accelerate training. PET image reconstruction involves complex physical corrections including attenuation, random and scatter corrections, which complicate the noise pattern in the reconstructed PET image. Therefore, there are not only differences in the noise pattern, but also potential differences in the image structure between LDPET and FDPET (The major difference is the noise pattern though). In LDPET denoising, we treated the different noise patterns as features, and the denoising process as the feature transfer. When we trained CycleGAN model in the unsupervised learning fashion, some artifacts of missing structures are observed in about 7% of images. 

So, the additional supervised learning loss effectively guides the transformation of the model mainly focusing on noise patterns. (More details can be found in Section “Ablation Study”, Line 251-263)

This claim is unproven: “the proposed method is better at preserving image details than the other two methods”. The samples shown in figures 7 and 8 seem to support this claim, but a quantitative justification should be given with the proper metrics. Indeed, the NRMSE, SSIM, and PSNR are not adequate perceptual metrics (https://arxiv.org/abs/1801.03924). A perceptual loss (LPIPS) or a mean opinion score measure (MOS) could be used to support this statement.

Thanks for your suggestion, we added a LPIPS comparison plot for S-CycleGAN, RED-CNN and 3D-cGAN. Our proposed model, S-CycleGAN, could obtain the best perception score. (Figure 9)

Miscellaneous remarks:

- Typo in the abstract: “S-CylceGAN”

- Typo line 78: “supervise”

We appreciate the reviewer for helping us to improve grammar and we have revised the manuscript appropriately. 

Reviewer #2: This paper investigates a supervised, cycle-consistent, conditional GAN to map low-dose PET brain images to full-dose images. Reported performance is promising compared to other deep learning frameworks. However, the study has two main limitations that need to be addressed before possible publication.

* A training set (99 samples) and a test set (10 samples) were defined. However, no validation set is mentioned, which is a severe limitation. In particular, it is not clear how the numerous hyperparameters (the alpha, beta and gamma loss weights, the learning rate, Adam optimizer's beta1 and beta2 parameters, the patch size, network hyperparameters, etc.) were set: if the number of hyperparameters is close to the number of test samples (10), then the selection criterion for these hyperparameters is very critical. Also, it is not clear how the stopping criterion was defined. Training is generally stopped when the validation loss is minimal. When is it stopped if no validation set was defined? When the test loss is minimal? Given the small number of samples (109), I recommend to follow a cross-validation strategy on the 99 development samples: the value of hyperparameters (including the number of training epochs) could be defined as those maximizing the cross-validation score. A final model would then be trained on the entire development set using the optimal hyperparameter values.

Thanks for your suggestion. Since we didn’t have plenty of clinical PET images, we cropped image patches from the original PET images to augment the datasets for both training and testing. Actually, there were 136704 and 15360 patches (i.e. 89 and 9 patients) for training and validation though we didn’t include those details in the previous manuscript (See Section “Datasets”). 

The hyperparameters basically were derived from the original CycleGAN paper. As for the parameter, γ, it was determined by experiments in order to get a trade-off between the noise and the bias of SUV in the lesion regions.

* Each of the four training losses (Adversarial Loss, Cycle Consistency Loss, Identity Loss, Supervised Learning Loss) seem relevant. However, it is not clear how each of them contribute to the overall performance. Note that the alpha, beta and gamma weights are not reliable indicators in themselves. The added value of this paper would be much higher if the impact of each loss function was analyzed independently: this could be done by retraining the system without the loss function to analyze.

We made an ablation study to demonstrate our design, and results have been included in the manuscript. (Line 251 – 263)

Minor comments are listed below:

* Other papers address the same task using CycleGANs (for instance "Yang Lei et al. Low dose PET imaging with CT-aided cycle-consistent adversarial networks, Proc. SPIE 11312, Medical Imaging 2020: Physics of Medical Imaging, 1131247 (16 March 2020); https://doi.org/10.1117/12.2549386"). Differences with these papers should be highlighted.

* The expression of the Wasserstein distance should be provided.

* Training details for RED-CNN and 3D-cGAN solution should also be provided. Moreover, methodological differences between these solutions and the proposed solution should be better highlighted.

* Why providing hardware details (NVIDA TITAN GTX GPU) if no computation times are given? Training times and inference times should be provided.

The references, the expression of Wasserstein distance, training details for RED-CNN, 3D-cGAN, and training and inference time information have been appropriately added. 

Typos:

* applicaions -> applications

* an Machine Learning -> a Machine Learning

We appreciate the reviewer for helping us to improve grammar and we have revised the manuscript appropriately. 

Reference:

Harms, J., Lei, Y., Wang, T., Zhang, R., Zhou, J., Tang, X., Curran, W.J., Liu, T., Yang, X., 2019. Paired cycle-gan-based image correction for quantitative cone-beam computed tomography. Medical Physics 46, 3998–4009. URL: https://onlinelibrary.wiley.com/doi/abs/

10.1002/mp.13656, doi:10.1002/mp.13656.

Data Availability Statement

The data are available within the paper and on Figshare:

https://figshare.com/articles/dataset/LDPET/12686099

Funding Statement

MinFound Medical Systems Co., Ltd provided support for this study in the form of salaries for authors HY, LZ, SG, XW, and YW. The specific roles of these authors are articulated in the ‘author contributions’ section. The funders played a role in study design, data collection and analysis, decision to publish, or preparation of the manuscript. No additional external funding was received for this study.

Competing Interests Statement

The authors have read the journal's policy and have the following competing interests: HY, LZ, SG, XW, and YW are paid employees of MinFound Medical Systems Co., Ltd. There are no patents, products in development or marketed products associated with this research to declare. This does not alter our adherence to PLOS ONE policies on sharing data and materials.

---

## [Decision Letter · Decision Letter 1]

18 Aug 2020

Study of low-dose PET image recovery using supervised learning with CycleGAN

PONE-D-19-34960R1

Dear Dr. Ye,

We’re pleased to inform you that your manuscript has been judged scientifically suitable for publication and will be formally accepted for publication once it meets all outstanding technical requirements.

Kind regards,

Mathieu Hatt, MSc, PhD, HDR

Academic Editor

PLOS ONE

Additional Editor Comments (optional):

Only a few minor typos need to be corrected when preparing the final version.

Reviewers' comments:

Reviewer's Responses to Questions

**Comments to the Author**

1. If the authors have adequately addressed your comments raised in a previous round of review and you feel that this manuscript is now acceptable for publication, you may indicate that here to bypass the “Comments to the Author” section, enter your conflict of interest statement in the “Confidential to Editor” section, and submit your "Accept" recommendation.

Reviewer #1: All comments have been addressed

Reviewer #2: All comments have been addressed

2. Is the manuscript technically sound, and do the data support the conclusions?

Reviewer #1: Yes

Reviewer #2: Yes

3. Has the statistical analysis been performed appropriately and rigorously? 

Reviewer #1: Yes

Reviewer #2: Yes

4. Have the authors made all data underlying the findings in their manuscript fully available?

Reviewer #1: Yes

Reviewer #2: Yes

5. Is the manuscript presented in an intelligible fashion and written in standard English?

Reviewer #1: Yes

Reviewer #2: Yes

6. Review Comments to the Author

Reviewer #1: All concerns and weaknesses pointed out during the first round of peer review have been adequately addressed.

Typos:

- "CylcleGAN" line 253

- "Cycle-consistence loss" line 258 - "Cycle-consistency loss"

- "All experiments were conducted using the Keras with Tensorflow backend" line 188 - remove "the" and cite Keras/Tensorflow (https://keras.io/getting_started/faq/#how-should-i-cite-keras)

Reviewer #2: (No Response)

7. PLOS authors have the option to publish the peer review history of their article (what does this mean?). If published, this will include your full peer review and any attached files.

Reviewer #1: No

Reviewer #2: No

---

## [Editor Report · Acceptance letter]

24 Aug 2020

PONE-D-19-34960R1 

Study of low-dose PET image recovery using supervised learning with CycleGAN 

Dear Dr. Ye:

I'm pleased to inform you that your manuscript has been deemed suitable for publication in PLOS ONE. Congratulations! Your manuscript is now with our production department. 

Kind regards, 

on behalf of

Dr. Mathieu Hatt 

Academic Editor

PLOS ONE